# Exploring the Sensory Properties and Preferences of Fruit Wines Based on an Online Survey and Partial Projective Mapping

**DOI:** 10.3390/foods12091844

**Published:** 2023-04-29

**Authors:** Yuxuan Zhu, Qingyu Su, Jingfang Jiao, Niina Kelanne, Maaria Kortesniemi, Xiaoqing Xu, Baoqing Zhu, Oskar Laaksonen

**Affiliations:** 1Beijing Key Laboratory of Forestry Food Processing and Safety, Department of Food Science, College of Biological Sciences and Biotechnology, Beijing Forestry University, Beijing 100083, China; 2Food Sciences, Department of Life Technologies, University of Turku, 20500 Turku, Finland; niina.m.kelanne@utu.fi (N.K.); mkkort@utu.fi (M.K.)

**Keywords:** berry materials, consumers, consumption frequency, liking, sensory quality, online questionnaire

## Abstract

Non-grapefruits with unique sensory properties and potential health benefits provide added value to fruit wine production. This study aimed to explore consumers’ fruit wine preferences and descriptors for the varied fruit wines. First, 234 consumers participated in an online survey concerning their preferences for different wines (grape, blueberry, hawthorn, goji, *Rosa roxburghii*, and apricot). In addition, their attitudes towards general health interests, food neophobia, alcoholic drinks, and sweetness were collected. Grape wine and blueberry wine were the most favored wines, and goji wine was the least liked fruit wine sample. Moreover, 89 consumers were invited to evaluate 10 commercial fruit wines by using partial projective mapping based on appearance, aroma, and flavor (including taste and mouthfeel) to obtain a comprehensive sensory characterization. Multifactor analysis results showed that consumers could differentiate the fruit wines. Participants preferred fruit wines with “sweet”, “sour”, and “balanced fragrance”, whereas “bitter”, “astringent”, “deep appearance”, and “medicinal fragrance” were not preferred. Attitudes toward health, food neophobia, alcohol, and sweetness had less influence than taste and aroma (sensory attributes) on the preferences for fruit wine products. More frequent self-reported wine usage resulted in higher consumption frequency and liking ratings compared to non-users. Overall, the main factors influencing consumer preference for fruit wines were the sensory characteristics of the products, especially the taste.

## 1. Introduction

Fruit wines are medium-alcohol beverages (alcohol content: 5%vol~15%vol) made from non-grapefruit material. Fruit wines can be often favored by young and middle-aged consumers, especially young women [1]. This is consistent with increasing demand by consumers for fruit wines showing great development potential for fruit wine industry [1]. China is the world’s largest fruit producer, resulting in huge waste every year [2]. Winemaking can increase the added value of fruits for consumption and solve the problem of imbalance in fruit production and marketing [3]. Non-grape wines, with the exception of cider, account for a negligible share of the global wine market [4]. Similarly, the consumption positioning of fruit wine products is ambiguous, which greatly limits the long-term development of fruit wines in the Chinese market. While grape wines dominate the current wine market, an increasing number of producers have utilized several non-grapefruits to produce fermented wines to meet the diversified needs of consumers [5].

Alcohol content and sugar content are among the key determinants of consumer preference for alcoholic beverages [6,7]. Sensory analysis of fruit wines generally includes evaluation of appearance, aroma, and taste [8]. The sensory methods can be utilized in every step of wine production and final wine [9]. The sensory quality of various locally important fruit wines has been reported, including bog bilberry (*Vaccinium uliginosum* L.) wines [10], jackfruit (*Artocarpus heterophyllus*) wines [11], gabiroba (*Campomanesia pubescens*) wines [12], and cagaita (*Eugenia dysenterica*) wines [13]. Consumers are increasingly attracted to fruit wines due to their potentially low alcohol content, varying aroma profiles, and richness in various bioactive compounds compared to common grape wines [14,15]. Although grape wine is more prevalent in the global market and has more consumers, a survey indicated that young consumers may likely be attracted to novel fruit wines; thus, fruit wines have great potential for commercialization while simultaneously addressing the huge waste of fruit resources [16].

Healthiness of food or beverage may affect consumers’ consumption intention. Studies have shown that fruit and berry products are enriched with phenolic compounds and other antioxidants originating from the raw material [5,17,18]. According to Parpinello et al. [19], whether a beverage is good for health will affect consumers’ preference and acceptance: consumers with more interest in healthiness of foods are more likely to accept healthy but sour drinks. Furthermore, when consumers were informed about the anthocyanin content of the juices, they were more likely to buy samples highest in anthocyanins and less likely to buy juices lowest in anthocyanins [20]. The food neophobia scale (FNS) denotes “reluctance to eat and/or avoidance of novel foods” [21]. Although the Chinese fruit wine industry has experienced decades of development since its establishment, it is still a relatively small product category, e.g., relatively novel drinks for consumers in the broad market. Therefore, it is valuable to investigate whether consumers’ consumption of fruit wine is affected by food neophobia or other attitudinal features.

Sensory characteristics of wine can be considered as one of the key factors in the acceptance of the wince. The quality of wine is defined by several factors, including aroma, appearance, mouthfeel, and taste [22]. A study has shown that the appearance of wine affects aroma perception, whereas the aroma, in turn, affects flavor perception [8]. Due to the interactions among different sensory attributes, multisensory analytical methods are required. Projective mapping uses a projective approach to generate two-dimensional perceptual maps, asking individual assessors to place products on the space themselves based on their perceived similarities and differences [23]. Partial projective mapping is a sensory method that is used to evaluate one sensory modality (flavor, texture, aroma, or appearance) [24]. 

This study aimed to explore how consumers recognize the various sensory characteristics of fruit wines and their preferences. This study is also a continuation of our previous research [10], expanding the range of fruit wines. Since there are still few actual consumers of fruit wines, we invited adults to participate in the survey and divided them into “users” and “non-users” according to their self-reported consumption frequency. Participants were first invited to complete an online questionnaire survey, and the study continued with the sensory evaluation of different fruit wines by using a rapid sensory analysis method named Partial Projective Mapping [25]. Meanwhile, the effect of general health interest, food neophobia, and attitudes towards alcoholic drinks and sweetness on the consumers’ preferences and acceptances were analyzed. The results of this study will enable to better understand how consumers in China perceive the sensory characteristics of fruit wine products found in the market, thus guiding for the industry to improve the sensory quality of their fruit wine products.

## 2. Materials and Methods

### 2.1. Wine Samples

A total of 10 commercial fruit wines from 9 different Chinese manufacturers, made from 4 fruit ingredients, were selected and purchased online, including B1 (blueberry wine, Beiyushidai Company, Yichun, China), B2 (blueberry wine, Beiyushidai Company, Yichun, China), B3 (blueberry wine, Yicunshanye Company, Yichun, China), B4 (blueberry wine, Zifei Company, Qingdao, China), A (apricot wine, Zhuoyaxuan Company, Jinan, China), H1 (hawthorn wine, Daobaisui Company, Nanning, China), H2 (hawthorn wine, Lvti Company, Panjin, China), H3 (hawthorn wine, Shengbali Company, Qingzhou, China), R (*Rosa roxburghii* wine, Yunshangcilihua Company, Guanling Autonomous County, China), and G (goji berry wine, Ningxiahong Company, Yinchuan, China). The types of fruit wines and their alcoholic content are shown in Appendix A. All fruit wines were evaluated at room temperature.

### 2.2. Participants

The protocols (including ethical guidelines) of the sensory studies were approved by the College of Humanities and Social Sciences of Beijing Forestry University. At the same time, participants were informed about the usage of data collected in this study (aimed for a scientific publication and how personal data was to be handled). An online questionnaire was distributed to students and staffs at Beijing Forestry University and publicly advertised at the university. A total of 238 questionnaires were received, but one had incomplete information, and three were duplicated, resulting in a final valid questionnaire of 234. A total of 148 of the 234 participants were female, accounting for 63% of the total, and 86 were male, accounting for 37% of the total. The age of the questionnaire participants ranged from 18–60 years old, with 11% of participants aged 18–20, 78% aged 20–29, 8% aged 30–39, and 3% aged 40 and above. Among the participants were 197 students, 11 teachers, and 26 others (company employees, retired, etc.). Similar to the online survey, we recruited 89 participants for the sensory experiment. 

### 2.3. Online Questionnaire

An online questionnaire was conducted before the sensory experiment to understand consumption of fruit wines and attitudes toward them. The questionnaire was divided into three parts. The first part was to collect the general demographic information: gender, age, occupation, education level, etc. Six types of fruit wines (grape wine, blueberry wine, hawthorn wine, goji berry wine, *R. roxburghii* wine, and apricot wine) were selected for the survey, and consumers were asked to rate them on a 5-point scale (liking: 1 = “dislike very much” to 5 = “like very much”; familiarity: 1 = “very unfamiliar” to 5 = “very familiar”; frequency of consumption: 1 = never, 2 = several times a year, 3 = 1–3 times a month, 4 = 2–5 times a week, 5 = daily). It also includes the following multiple-choice questions: “How often do you drink alcoholic beverages?”; “What is your acceptable alcohol content range?”; “When buying fruit wine, what aspects of fruit wine do you value more?”. The third part of the survey was to investigate the consumer’s attitudes on health, food neophobia, alcoholic drinks, and sweetness. Questions were presented and rated using a 7-point scale for respondents to indicate their extent of agreement (1 = “strongly disagree” to 7 = “strongly agree”). General Heath Interest (GHI) questions with eight statements from Roininen et al. [26] were adopted in this study. Food Neophobia Scale (FNS) [21], attitudes toward alcoholic drinks, and attitudes toward sweetness each contain six statements in this study. Thus, the potential range of the GHI scale scores was 8–56. The potential range of the FNS, attitudes toward alcoholic drinks, and attitudes toward sweetness was 6–42.

### 2.4. Sensory Test

Participants were first informed about the nature of the test (related to ethical issues): what and how the data was collected, and how we intended to use it (anonymity, scientific publication, etc.). In addition, they were informed that they could cancel their participation at any time. The fruit wine samples were confirmed by experienced wine professionals to be free of contamination and were ensured to be freshly opened. In the sensory test, the same Projective mapping (PM) method was used as described by Esmerino et al. [27]. The samples (*n* = 10) were all prepared according to the same procedure. They were presented in international standard tasting glasses simultaneously with a sample size of 30 mL. Samples were marked with a three-digit random code on each glass, and each sample was presented to the participants. During the sensory test, water and soda crackers were provided to each participant for palate cleansing. The sensory test was conducted at room temperature. Samples were poured and presented 10 min before the sensory analysis. The glasses were covered with paper caps and would not be opened by the participants until tasting.

To evaluate the samples, participants were asked to evaluate all the samples in 3 sessions and were asked to place the samples in two-dimensional space according to the similarities or differences in sensory characteristics [25]. First, they were asked to observe the appearance of all the samples, then smell all the aromas, and finally, taste all the samples. They were instructed to chew a piece of cracker and rinse their mouth with water between samples for palate cleansing. Afterwards, participants were asked to give a 9-point preference score for each sample (1 = “extremely dislike” to 9 = “extremely like”). Compusense software V19.0.1 (Compusense Inc., Guelph, ON, Canada) was used to collect data from the sensory experiments. The sensory experiments were conducted in Beijing Forestry University. Participants evaluated samples in a quiet, clean laboratory where each person had enough space to ensure that they were not disturbed.

### 2.5. Qualitative and Quantitative Analysis of Sugars and Organic acids with GC-FID

Standard compounds: d-(−)-glucose and d-(−)-fructose were purchased from Merck (Darmstadt, Germany). Succinic acid, malic acid, tartaric acid, quinic acid, xylitol, and galacturonic acid were purchased from Sigma–Aldrich (St. Louis, MO, USA). Sucrose and citric acid (monohydrate) were purchased from J.T. Baker Chemicals (Leuven, Belgium).

The sugars and organic acids were analyzed with a GC (GC-2010Plus, Shimadzu corp., Kyoto, Japan) equipped with a flame ionization detector (FID) as trimethylsilyl (TMS; Tri-Sil reagent, hexamethyldisilazane:trimethylchlorosilane:pyridine, 2:1:10, Thermo Scientific, Pierce Biotechnology, Rockford, IL, USA) derivatives as described before by Kelanne et al. [28], with slight modifications. External standards (succinic acid, citric acid, fructose, quinic acid, glucose, galacturonic acid, glycerol, sorbitol, malic acid, ascorbic acid, and sucrose, all 5 g/L) were used for identification and quantification of the main sugars and organic acids in fruit wine samples. Xylitol and tartaric acid were used as internal standards for quantification of sugars and organic acids, respectively. An aliquot portion of 250 µL of each sample and both internal standards were diluted to 5 mL and filtered with a regenerated cellulose syringe filter (0.45 µm). An aliquot portion of 300 µL of the filtrate was pipetted to an autosampler bottle and evaporated to dryness at 50 °C under nitrogen flow. The samples were stored at a desiccator until analysis, but at least overnight. For TMS derivatization, 500 µL of Tri-Sil reagent was added to dry samples, mixed vigorously for 5 min, and incubated for 30 min at 60 °C. Separation of derived compounds was carried out with a Supelco Simplicity-1 fused silica column (30 m × 0.25 mm i.d. × 0.25 μm df, Supelco, Bellefonte, PA, USA). The oven was temperature-programmed from 150 °C (hold = 2 min) to 210 °C with a constant ramp of 3 °C/min, then to 275 °C (hold = 5 min) with a constant ramp of 40 °C/min. The injector temperature was set to 210 °C, and split injection was applied with a split ratio of 1:15. Helium was used as a carrier gas with a linear velocity of 44.8 cm/s (constant flow). The temperature of the FID was set to 290 °C. All samples were analyzed in triplicate.

### 2.6. Data Analysis

TwoStep Cluster Analysis was used for population grouping in GHI (two groups), FNS (three groups), attitudes toward alcoholic drinks (three groups), and attitudes toward sweetness (three groups). The differences in liking, familiarity, and frequency of consumption among these groups were also analyzed by ANOVA using Tukey’s post hoc test. Differences in liking, familiarity, and frequency of consumption among samples were examined using ANOVA with the sample gender as a fixed factor. Differences in liking and familiarity among users and non-users were used independent samples t-test. Differences in the sugar and organic acid contents were analyzed with one-way ANOVA using Tukey’s post hoc test. Three-way ANOVA models (gender, GHI, FNS, attitudes toward alcoholic drinks, and attitudes toward sweetness as fixed factors) were used to assess differences in liking among subgroups within the sample, and significance was further tested using Tukey’s post hoc test. Compusense software was used to record the coordinate locations (*x* and *y* coordinates) of each participant’s product map. The sensory attribute descriptors for each sample provided by the experiment participants were summarized. When summarizing the sample sensory descriptors, all words were ranked and deleted enjoyment terms (e.g., good taste, dislike, etc.), similar descriptors in the filtered descriptor list were combined, and the frequency of descriptor occurrences was counted. 

All intensities of attributes (e.g., slightly sweet, sweet, and very sweet) were treated as separate attributes [29]. Multivariate analysis was used to analyze the results of the sensory experiments. Multiple Factor Analysis (MFA) was performed on PM dataset with coordinates given by respondents as two active groups using the ‘FactoMineR’ package with R version 4.0.3 (R Foundation, Vienna, Austria) [30]. An individual factor map of the first two dimensions was obtained to illustrate the product space given by consumers. Descriptors were used as a supplementary group in the MFA map to assist interpretation of the product map. A frequency table of descriptors for each sensory attribute of each sample was visualized in word clouds with the font of the descriptors proportional to the frequency of mentions by R package ‘wordcloud’ [31] for each product. PCA was used to analyze the relationship between sensory outcomes and preferences using the R package ‘factoextra’. Liking segments were explored via k-means clustering. Barplots were conducted by R package ‘ggplot2’ [32]. All ANOVAs and Cluster Analysis were analyzed by IBM SPSS Statistics 22.0 (IBM Corporation, Armonk, NY, USA).

## 3. Results

### 3.1. Online Questionnaire on Liking, Familiarity, and Consumption Frequency

Cluster analysis was used to classify respondents into ‘users’ and ‘non-users’ according to their reported frequency of consumption. The data of the questionnaire were analyzed later using independent samples t-test, which revealed differences in preference and familiarity with different fruit wine types (Table 1). Respondents in this study were relatively infrequent drinkers of fruit wines. The participants were divided into users (consumption frequency score ≥ 10) and non-users (consumption frequency score < 10). Among the different groups, grape wine and blueberry wine were the most favored fruit wines, with average liking scores of 3.64 and 3.62 (out of 5 points). Expectedly, the grape wine was one of the most familiar wines for consumers. Although consumers were aware of Goji berry wine, the preference score was below. Apricot wine and *R. roxburghii* wine were the least familiar, with scores of 2.08 and 2.01, respectively, and these two fruit wines were also the least commonly consumed fruit wines. Gender showed variability of preference, familiarity, and frequency of consumption in only grape wine and hawthorn wine: female consumers preferred both fruit wines compared to male consumers (Appendix A). However, male consumers gave higher scores for familiarity with these two fruit wines.

The factors consumers were most concerned about when purchasing fruit wine products were taste, aroma, alcohol content, and appearance (Appendix A). The range of alcohol content with the highest consumer acceptance was 8–15% (Appendix A). Samples with alcohol concentrations within this range were selected for subsequent sensory experiments.

### 3.2. Impact of GHI, FNS, Attitudes for Alcoholic Drinks, and Attitudes for Sweetness

Based on participants’ scores (Appendix A) for GHI, FNS, attitudes for alcoholic drinks, and attitudes for sweetness, the consumers were all divided into different groups: GHI1, the least health interested (*n* = 126, GHI score ≤ 38); GHI2, the most interested (*n* = 108, GHI score > 38); FNS1, the least neophobic (*n* = 85, FNS score ≤ 23); FNS2 (*n* = 75, 23 < FNS score ≤ 26); FNS3, the most neophobic (*n* = 83, FNS score > 26); Alcohol 1, the negative attitude to alcohol (*n* = 71, Alcohol score ≤ 20); Alcohol 2 (*n* = 80, 20 < Alcohol score ≤ 25); Alcohol 3, the positive attitude (*n* = 83, Alcohol score > 25); Sweet1, the least craving for sweet foods (*n* = 73, Sweet score ≤ 22); Sweet 2 (*n* = 84, 22 < Sweet score ≤ 27); and Sweet 3, the most craving for sweet foods (*n* = 77, Sweet score > 27). The mean scores of likings, familiarity, and usage ratings of the above groups were calculated separately and analyzed by ANOVA (Table 2).

There were no significant differences in the effects of FNS on liking, familiarity, and usage rating. The GHI group showed significant differences in liking, with GHI1 showing a higher preference for all six fruit wines. Attitudes toward alcoholic drinks significantly affect liking, familiarity, and usage ratings. Alcohol 3 (the positive attitude) had the highest liking and familiarity scores for grape wine, apricot wine, and *R. roxburghii* wine. Alcohol 3 also has the highest usage rating of fruit wine, which matches our predictions. Overall, those with the least sweet tooth (Sweet 1) had a higher liking and familiarity with all six types of fruit wines, with grape wine and goji berry wine being the most significant. Significant differences in the usage frequency were observed only in apricot wine.

### 3.3. Sugars and Organic Acids in Fruit Wines

The content of sugars and organic acids varied widely among different fruit wines and among fruit wines made from the same raw material (Table 3). The highest total sugar content was observed in hawthorn wine (H2; 329 g/L) and the lowest in Goji berry wine (8.5 g/L). In addition, hawthorn wine H2 had the highest sugar-to-acid ratio (S/A; 33.5). The lowest S/A was observed in the blueberry wine B4 (0.2). B4 was the only sample that contained higher acids than sugars. Glycerol was the main sugar in two blueberry wine samples (B3 and B4). The sweetness of glycerol was 80 % of the sweetness of sucrose. In addition, sorbitol was only detected in the hawthorn wines (2.1–10.6 g/L). The highest organic acid content was observed in the blueberry wine B1 (14.9 g/L) and the lowest in the Goji berry wine (2.4 g/L). Ascorbic acid was detected only in the *R. roxburghii* wine.

### 3.4. Sensory Characterization of Fruit Wines

A total of 89 participants in the PM sensory experiment provided 146 descriptors to aid in demonstrating their sample distribution on the map. The PM results visually demonstrated the association between the samples and their descriptors. Of these, participants used 43 appearance descriptors to describe the 10 fruit wines (Figure 1) with the first two dimensions explaining 47.04% and 15.82% of the total data variance, respectively. In total, they explained 62.86% of the variance and were able to better explain the differences among the samples. The first dimension distinguishes among four blueberry wines (B1, B2, B3, and B4), two hawthorn wines (H1 and H3), and *R. roxburghii* wine (R). Additionally, the second dimension distinguishes among hawthorn wine (H2), apricot wine (A) and goji berry wine (G). B3 and B4 were characterized by crimson, deep, and turbid; B1 and B2 were characterized by purple, fuchsia, and deep purple; H1 and R were characterized by orange-yellow and clear; H3 was characterized by yellow and light; H2 was characterized by reddish-brown; A was characterized by orange-red; and G was characterized by light yellow. Sample appearance word cloud (Figure 1) was drawn based on appearance descriptors provided by participants in the test, which was consistent with the sample features shown in the PM map, indicating that participants were able to distinguish the similarity and difference of appearance among different samples.

Participants used 49 aroma descriptors to describe fruit wines (Figure 2). The first and second dimensions in PM explained 34.11% of the total data variance. Four blueberry wines and an apricot wine (B1, B2, B3, B4, and A) were distinguished from Goji berry wine (G) in the first dimension, and *R. roxburghii* wine (R) was distinguished from three hawthorn wines (H1, H2, and H3) in the second dimension. The PM map shows that consumers were essentially able to distinguish several wines based on similarities and differences in aroma. Participants who participated in the sensory experiment were able to identify the main aroma characteristics of the different fruit wines despite being untrained, as demonstrated by previous studies [33]. The main aroma characteristics of samples B1, B2, B3, and B4 were blueberry and grape. Sample G was dominated by fresh fragrance. Sample A was perceived with a pungent aroma, and samples H1, H2, and H3 were considered with hawthorn aroma as the key characteristic. Sample R showed an unbalanced fragrance and herbal aroma.

A total of 54 flavor descriptors were perceived for fruit wines. The first two dimensions in PM explained 35.92% of the total variance (Figure 3). A large variation was observed among fruit wine samples. The first dimension was mainly dominated by sweetness and sourness on the positive side and sourness and astringence on the negative side. The four blueberry wines (B1, B2, B3, and B4) samples and the three hawthorn wines (H1, H2, and H3) samples were distinguished by the first dimension. Blueberry wine samples were in the negative half-axis of the first dimension, and their taste characteristics were related to sour, bitter, and astringent. Hawthorn wine samples were in the positive half-axis of the first dimension. Their taste characteristics were mainly related to sweetness, sweet and sour, and fruity taste. This distinction may be caused by the difference in winemaking ingredients. Spicy and stimulating flavors were the main attributes of the positive half-axis of the second dimension. The taste characteristics of sample A were mainly related to stimulating and spicy taste; the characteristics of sample R were mainly related to bitterness and not sweet taste, while sample G was mainly related to the fresh taste and slight sweetness. At the same time, there were five words describing aftertaste in taste descriptors, such as sweet aftertaste, sour aftertaste, astringent aftertaste, bitter aftertaste, etc., indicating untrained consumers could experience aftertaste from different fruit wine samples.

Word clouds were created based on the descriptors provided by the participants in the sensory experiment (Appendix A). The word clouds graphically represent the fact that fruit wines with the same ingredients exhibit different appearances, aromas, and taste characteristics. The aroma characteristics of the hawthorn wine sample H2 were mainly hawthorn, while other aromas did not show prominently. The other hawthorn wine samples were different: H1 was mainly pungent, and H3 delivered hawthorn and fruity and apricot notes, which may be related to the different processing techniques. In terms of taste characteristics, blueberry wine sample B2 can be most perceived by consumers as sweet and sour, while samples B1, B2, and B3 were perceived as sour, astringent, and strongly alcoholic, which were related to the fact that sample B2 was the only sweet wine among the three. Consumers who participated in the sensory test, although untrained, were able to identify sensory characteristics that differed among the samples, demonstrating the reliability and reproducibility of the results. Additionally, the presence of alcohol (aroma) and alcoholic (taste) flavors in the aroma and taste descriptors of each wine was possibly because the participants were not trained.

By comparing the aroma descriptors given by consumers with the taste descriptors, they were more able to perceive and more willing to give taste-related descriptors. Therefore, the taste may be more likely to influence consumers’ preference and acceptance of fruit wines.

### 3.5. Liking of Fruit Wines in the Sensory Experiment

Liking scores of fruit wines are shown in Figure 4. The most preferred sample by consumers was H2 (a sweet hawthorn wine with 8% alcohol content), followed by sample G (a semi-dry goji berry wine with 7% alcohol content), and the least preferred by consumers was sample R (a sweet *R. roxburghii* wine with 12% alcohol level). To investigate the segmentation of the consumers based on their liking patterns of those fruit wines, k-means clustering was applied on the liking scores. K was set as 2. Results showed that consumers showed no clear segmentations on liking between the two groups. (Appendix A). Both groups of consumers demonstrate a similar dislike towards sample R and preference for sample H2.

To further investigate potential liking drivers of the fruit wines, PCA was conducted on the liking scores of 89 participants with the sensory descriptors from PM regressed on the biplot. It can be observed from the liking map of consumers that the first dimension explained 41.1% of the total data variance, and the second dimension contributed 11.3%. The biplot was largely stretched by samples H2 and R, which were the most and least liked products by the consumers. To further explore and visualize the potential sensory drivers of preference of the tested products, descriptors were regressed on the map with different thresholds of cos2 value being set. At cos2 ≥ 0.5 (Figure 5A), the sensory descriptors that performed a role in the placement of the samples were shown: samples H2 and G, for example, were associated with good taste, very sweet, sweet, and sour-sweet in the taste descriptor, and with balanced fragrance, very sweet (aroma), and sweet (aroma) in the aroma descriptor, and these positive sensory characteristics may have influenced the consumer’s high preference; sample R was associated with aroma dissonance, grassy and pungent in the aroma, and medicinal fragrance, aggressive, and unpleasant in the taste descriptors, and these negative sensory characteristics descriptors may have contributed to the low consumer preference for sample R. When cos2 ≥ 0.65 was set (Figure 5B), and the descriptors that were highly related with the liking difference of samples were clearer. In general, blueberry wines were associated with an astringent and sour taste, berry aroma, and deep color. The liking of hawthorn wines was generally linked with the sweet and sour taste. Medical and unbalanced aroma were highly associated with *R. roxburghii* wine. In summary, taste and aroma were key to the liking of the fruit wine products in this study.

## 4. Discussion

A total of 234 consumers participated in the online questionnaire survey. Respondents were mainly younger consumers (aged 18–40) as the adult consumers were the main group of wine consumers [34]. With research showing that 68% of young people (18–35 years old) say they would like to drink fruit wine in the future, the potential of the fruit wine market is huge [16]. Participants, who were more interested in health, gave higher liking ratings in general in comparison to the less interested. Regarding the possible healthiness of alcoholic beverages rich in fruit or berry material, the significant presence of ethanol makes them unhealthy despite of the possible healthy compounds derived from the raw material. Thus, the focus in beverage industry could be steered towards production of non-alcoholic fermented beverages and “wines” in order to promote healthiness of the fruit or berry material. In comparison to GHI, there were no similar findings among different FNS groups, suggesting that participants’ fear of new foods did not affect their preference for these fruit wines. In a previous study, participants who were more interested in health and were more neophobic preferred berries, especially the samples with lower preference scores [35]. Ristic et al. [36] developed the Wine Neophobia Scale (WNS) based on the FNS and demonstrated that the tool can be used to pre-screen and segment wine consumers. This facilitates product development for the wine industry and attracts consumers unfamiliar with wine. It may be useful for the fruit wine industry. The group with a positive attitude towards alcoholic beverages preferred more familiar and consumed these fruit wines more frequently. The participants preferring sweet did not like the wines very much, probably because they had a sour taste. 

Among the 234 participants in the online survey, there were 89 grouped as “users” and 145 “non-users”, who were differentiated by their self-reported frequency of fruit wine consumption. The “users” had higher preference and consumption frequency of wines. The frequency of consumption affects the participants’ preferences, but both users and non-users could distinguish the wine samples without much influence on the results of sensory tests [37]. Non-users had different expectations of fruit wines than users, and there were differences between their expectations and the real samples. For example, more than half of non-users would expect a fruit wine to taste close to the raw fruit, but different fruit types and fermentation strategies would result in an inconsistent taste experience. Users would not expect this based on their consumption experience [16]. This provides some help for fruit wines to attract non-users in the future. A shortcoming of this study is the selection of the “convenience consumer samples”, most of whom are school students. This may lead to a bias in the acceptability results with the “representative consumer samples” [38]. However, the actual consumers of these samples were very few and difficult to find enough of them. As this study supports the production of fruit/berry wines from locally important raw materials, consumer acceptance should be investigated with actual consumers of these products and more locally in the future studies.

A total of 89 untrained consumers were invited to perform a sensory evaluation on 10 different fruit wines using the rapid sensory analysis method of projective mapping. Projective mapping enabled the identification of the main sensory characteristics of the ten fruit wines in an efficient and informative way. Sensory maps in taste, aroma, and appearance were obtained, in which consumers positioned products based on their similarities and differences. Partial projective mapping provides more detailed information on wines and reduces the presence of hedonic attributes compared to global projective mapping [39]. Some researchers have used partial projective mapping to find differences in consumers’ flavor descriptions of white wines under different light sources [24]. These maps enabled a comparison among samples and the determination of their main sensory characteristics. The terms that participants use to describe samples could distinguish them, suggesting the validity of the vocabulary generated in the descriptive phase of projective mapping. As projective mapping was performed with untrained assessors, a few inaccurate and hedonic terms, such as ‘delicious’ and ‘nice odor’, were excluded from the data analysis. When experts and consumers evaluate the aromas of wines using projective mapping, they have different understandings of the vocabulary and distinguish among samples [40]. While experts tend to differentiate samples based on wine overall quality, consumers will be based on their own preferences [41]. However, the great majority of the terms used to describe the fruity wines were specific sensory descriptors. Although “naive consumers” were able to distinguish between samples, they were not as discriminating as experienced panelists and trained panelists [42]. More studies are needed in the future to specify and standardize the key sensory characteristics in each fruit wine using trained sensory panels. Moreover, future sensory consumer studies focusing on the acceptance should include more frequent users of these fruit wine products. 

The participants were able to identify the different varieties of fruity wines based on sensory characteristics. The close positioning of the samples of the same ingredients on the map indicated that fruit wines made from the same ingredient show more similarities, and this has been used in other studies on wine to determine the reproducibility of results [43]. White wines were separated by their variety in PM and ultra-flash profile (UFP) studies [44]; both the positioning of the identical samples and the separation of wines based on the variety indicates that the participants could distinguish differences in the wines presented to them. In this fruit wine study, the separation of different types of fruit wine showed a consistent result. This is the advantage of PM: the difference in sensory characteristics can be given in a short time. However, it is unable to distinguish subtle differences between samples and requires time to introduce the method to participants [45]. In contrast, CATA is a much easier to understand method [46] and has been shown to work in our previous studies [10]. PM is similar to UFP in that it produces a large number of undefined attributes that do not help to interpret the results. In the future, a table of attributes with definitions can be considered [44].

In general, the samples were separated based on the perceived taste, including sweetness, sourness, bitterness, and astringency. In the results, sweetness, sourness, and bitterness were opposite sides to each other on PCA. In addition, they were the most frequently used terms used to describe the fruit wine samples. In the literature, sweetness, bitterness, and fruity taste are all well-established attributes used to describe grape and fruit wines [47,48]. Two hawthorn wine samples (H1 and H2) had the highest sugar-to-acid ratios (Table 3), indicating them to be the sweetest samples, and they indeed correlated positively with “very sweet”, “sweet aftertaste”, and “sweet and sour” on PM (Figure 3) [49]. Goji berry wine had lower sugar-to-acid ratio than apricot and *R. roxburghii* wines, but it correlated positively more with “slightly sweet” than the other two wines, the reason of which may require further research. Volatile compounds are contributing to the fruity flavor, but they can also increase the sweetness perception of wine [50]. Blueberry wines correlated positively with sour and astringent attributes. These wines had the relatively highest contents of quinic acid compared to sugar content and moderate content of citric acid (Table 3), which have most likely caused sourness in these wines due to their sour taste properties. Our previous research showed that “fruity” and “floral” blueberry wines were preferred by consumers, with “spicy” and “herbal” flavors resulting in lower levels of preference [10]. The fruity flavor was often located near sweetness on the map. Furthermore, fruity is used to describe grape and fruit wines by consumers in the literature, and it could be characterized as a reliable and expected characterization for all wine types [51].

The aftertaste was mentioned frequently by the participants on the studied fruit wines, and it agrees with the study by Golia et al. [52] on red wine samples. In addition, Honoré-Chedozeau et al. [53] found that unfamiliar consumers without sensory training used more basic terms, such as a strong or light aftertaste, to describe wines. This result agreed with this study and indicated that consumers do notice and value the aftertaste of wine but usually do not describe the aftertaste. Consumers are more likely to indicate a presence or lack of aftertaste.

Through the PM session, the participants focused on the taste and aroma of the fruity wines. In the session, more attributes focused on the aroma and taste of the fruity wines; however, the participants used fewer descriptors to describe the appearance of the fruity wine. It was known that flavor (aroma and taste) had the most substantial influence on consumers’ overall food choices [54], and this might explain why the participants in this trial focused on the taste and aroma attributes of the fruity wine samples. The taste was important to the consumer acceptability of red wine and fruity wine [55] but was not identified as easily as appearance and aroma attributes. Untrained consumers might not be familiar with what the wine body was and, therefore, do not identify it when they were evaluating wine [56]. Consumer familiarity with the basics of wine or fruit wine should be increased before sensory experimentation. Lastly, the appearance of the fruity wine samples was largely ignored, but appearance, specifically the color of the wine, performs an essential role in the assessment of the wine [57]. Overall, the aroma and taste of the fruity wine was the most important aspect to the consumers, and the sweet, sour, and bitter attributes were the frequently identified terms by the participants.

The samples H2 and G in the sensory test were rated as the most-liked samples and not rated as the least-liked by anyone. The appealing sensory characteristics of them are clear and transparent in appearance, sweet, fruity, and rich aroma and sweet, sour, and astringent taste. The sensory characteristics of these positive terms may be the key factor for fruit wine to obtain consumer preferences. Sample R was clearly the most frequently rated as the least-liked sample. The unfamiliar flavor of the sample was a more important factor contributing to being the least liked. The pungent, incongruous aroma and sour, astringent, and bitter taste of the sample were also important factors for the unpopularity of sample R.

## 5. Conclusions

The result of this study found that un-trained consumers could distinguish different fruit wines based on their sensory characteristics. By comprehensively collecting the sensory characteristics (appearance, aroma, and taste) of the fruit wines during the PM experiment, the taste and aroma of fruit wines were most important to participants. In contrast, appearance is not their main consideration. Untrained consumers always used “sweet”, “sour”, and “bitter” to distinguish fruit wine samples. From the results of PM and liking scores, we found that the main factor that affected consumers’ preferences was the sensory characteristics of the fruit wine products, especially taste. In general, the consumers preferred fruit wines with sweetness, sourness, and balanced fragrance, such as the blueberry wine. They do not like “astringent”, “bitter”, “medicinal fragrance”, and “strong sour” fruit wines, which should be avoided in future production. However, more studies are needed using trained sensory panels to specify many of the sensory attributes used here by the untrained panelists. Consumer attitudes (health, neophobia, alcohol, and sweetness) have limited influence on fruit wine preferences, whereas familiarity may be one of the key factors. The familiarity, liking, and frequency of consumption of the 6 fruit wines were surveyed online among 234 participants. Frequency of consumption significantly influenced participants’ familiarity and liking of fruit wines. In contrast to online questionnaires, sensory testing provided detailed reasons for influencing consumer preferences. Study also supported the utilization of non-grapefruits in winemaking to help the fruit wine industry. Especially when using locally important crops, such as wild blueberries from the northern regions of China used in some samples in this study, thus helping to reduce the waste of perishable raw materials.

## Figures and Tables

**Figure 1 foods-12-01844-f001:**
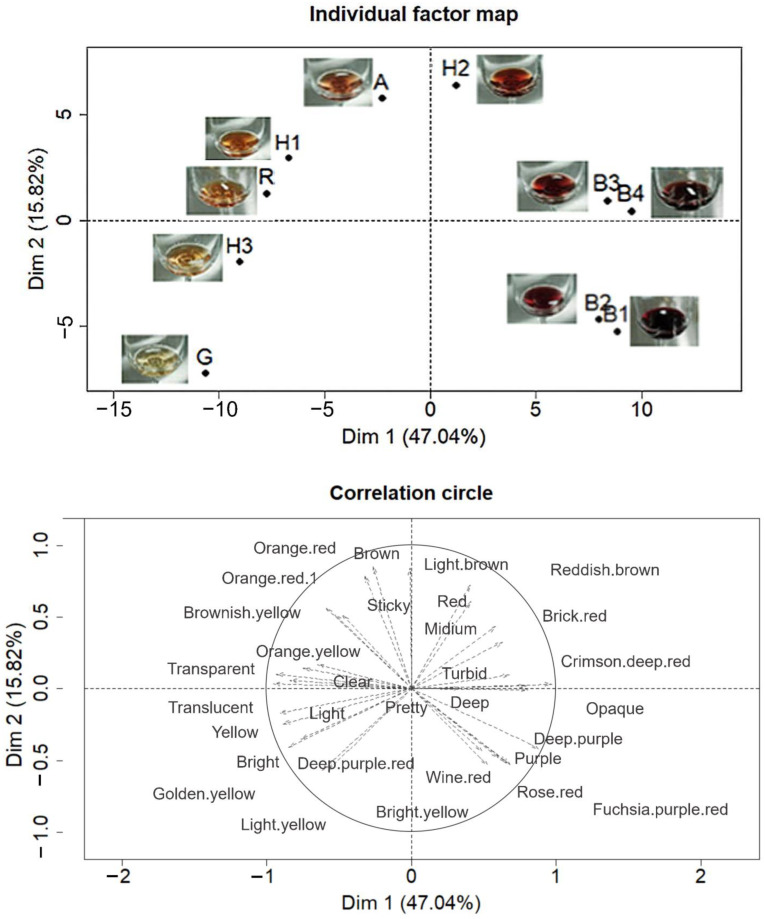
Representations of the ten fruit wine samples and the terms used to describe the samples for the first two dimensions of the multiple-factor analysis of the data from the Projective Mapping task with appearance.

**Figure 2 foods-12-01844-f002:**
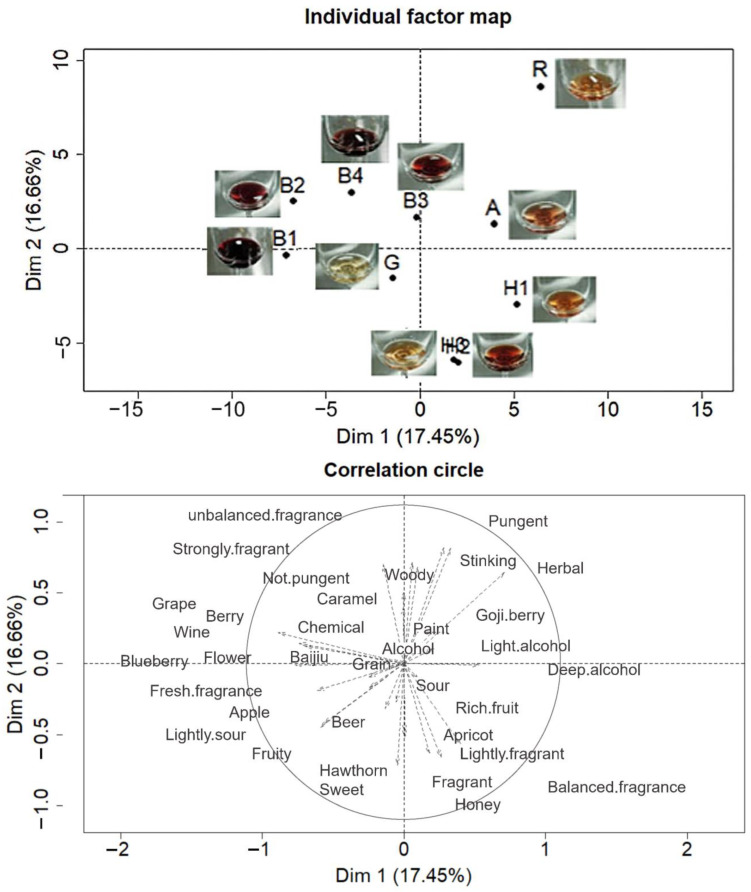
Representations of the ten fruit wine samples and the terms used to describe the samples for the first two dimensions of the multiple-factor analysis of the data from the Projective Mapping task with aroma.

**Figure 3 foods-12-01844-f003:**
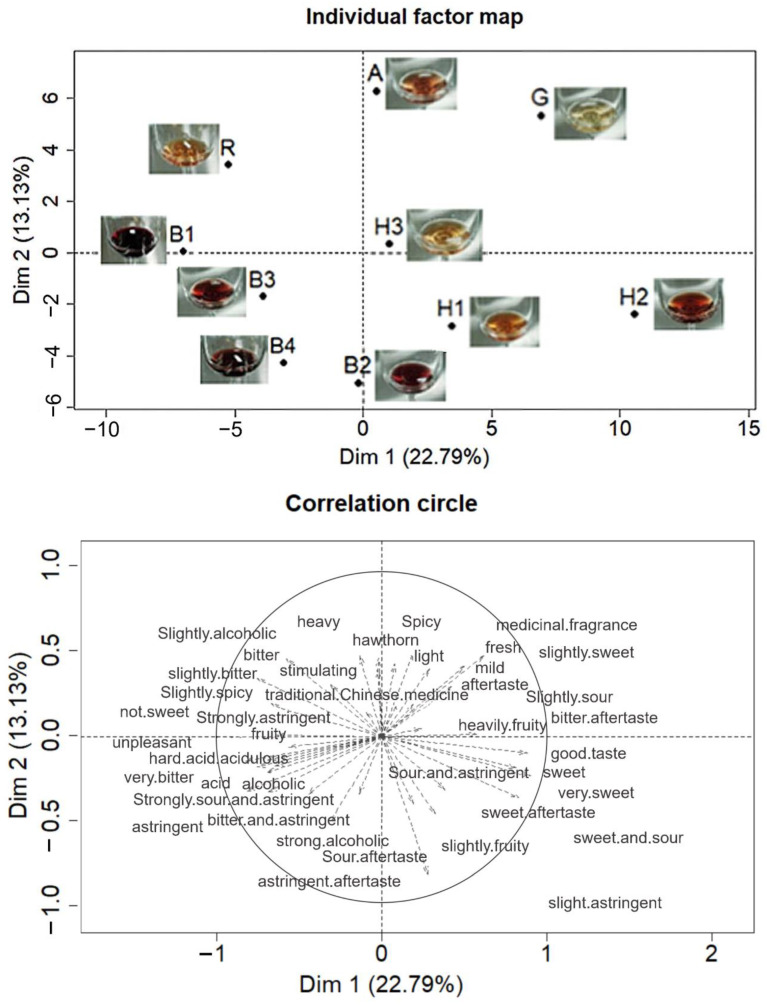
Representations of the ten fruit wine samples and the terms used to describe the samples for the first two dimensions of the multiple-factor analysis of the data from the Projective Mapping task with taste.

**Figure 4 foods-12-01844-f004:**
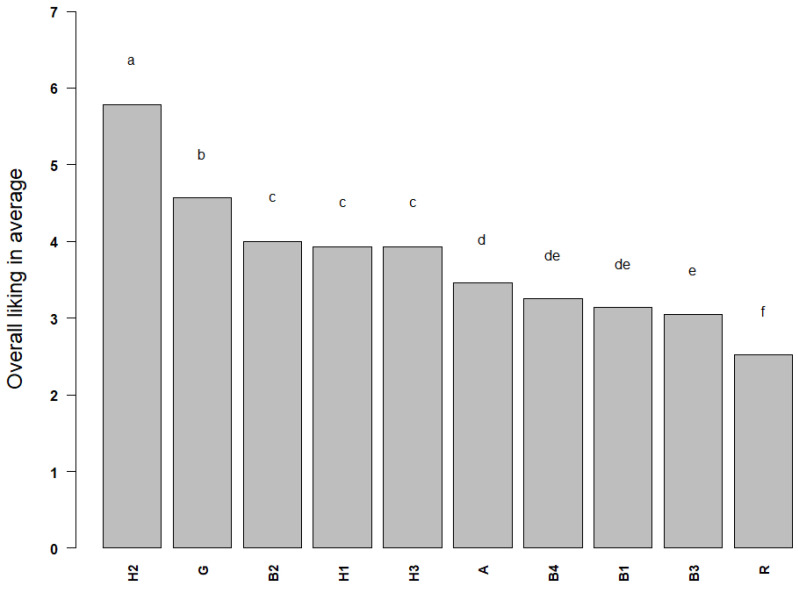
The overall average liking of fruit wine samples by the participants (*n* = 89) participating in the projective mapping test. Statistical differences were marked with a different letter (a–f) and were based on one-way ANOVA and Tukey’s test (*p* < 0.05).

**Figure 5 foods-12-01844-f005:**
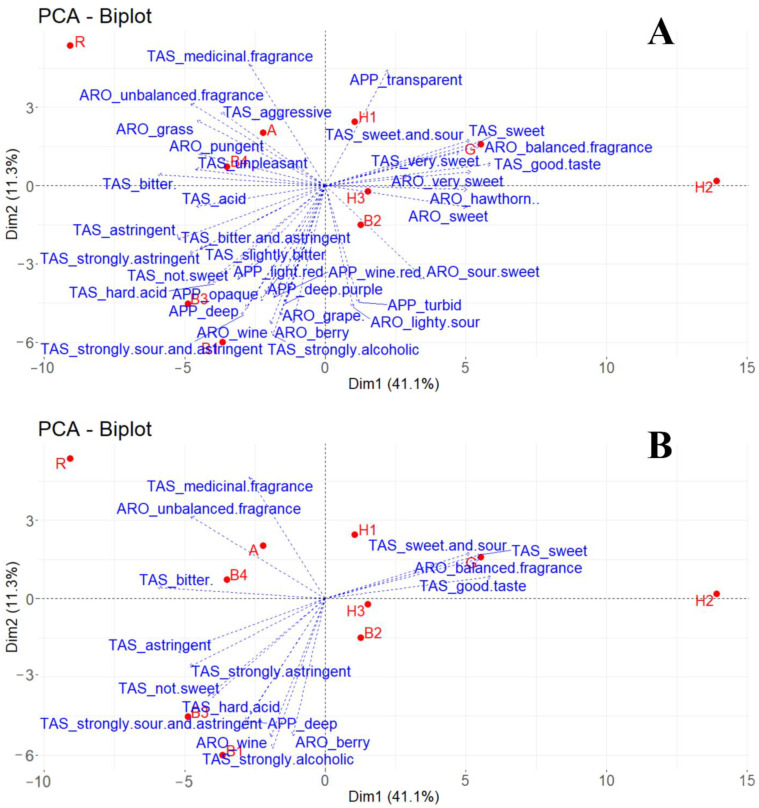
Principal component analysis (PCA) of liking map with descriptors (*n* = 146) from Projective mapping task. Select. var: cos2 ≥ 0.5 (**A**) and cos2 ≥ 0.65 (**B**) shown as the sensory descriptors that performed a role in the placement of the samples. If the cos2 is higher, this attribute is more important, and the attributes in (**B**) were the more important ones. APP (appearance), TAS (taste), and ARO (aroma).

**Table 1 foods-12-01844-t001:** Grouping according to frequency of consumption of each type of wines, means (±standard deviations) of liking and familiarity of different wine types in the online questionnaire (*n* = 234).

		Liking (1–5)					Familiarity (1–5)			
Fruit Wine type	Users (*n* = 89)	Non- Users (*n* = 145)	Significance	t	Overall Participants (*n* = 234)	Users (*n* = 89)	Non-Users (*n* = 145)	Significance	t	Overall Participants (*n* = 234)
Grape	3.91 ± 0.83	3.47 ± 0.87	***	3.83	3.64 ± 0.80	3.87 ± 0.88	3.35 ± 1.03	***	4.05	3.55 ± 1.01
Blueberry	3.93 ± 0.69	3.42 ± 0.85	***	5.05	3.62 ± 0.83	3.25 ± 0.82	2.38 ± 0.96	**	7.39	2.71 ± 1.00
Hawthorn	3.60 ± 0.95	3.20 ± 0.95	**	3.10	3.35 ± 0.97	3.11 ± 0.98	2.23 ± 0.95	***	6.84	2.56 ± 1.05
Goji berry	3.10 ± 1.00	2.84 ± 0.91	*	2.04	2.94 ± 0.95	3.09 ± 0.97	2.19 ± 0.97	***	6.90	2.53 ± 1.07
*Rosa roxburghii*	3.26 ± 0.89	2.88 ± 0.79	***	3.43	3.02 ± 0.85	2.33 ± 0.90	1.81 ± 0.70	***	4.58	2.01 ± 0.82
Apricot	3.43 ± 0.78	2.97 ± 0.84	***	4.20	3.14 ± 0.84	2.42 ± 0.90	1.88 ± 0.73	***	4.76	2.08 ± 0.84

Users (total consumption frequency score of six fruit wines ≥ 10) and non-users (total consumption frequency score of six fruit wines < 10). Independent samples t-test was used to check the difference between users and non-users. ‘*’ represents *p* < 0.05, ‘**’ represents *p* < 0.01, and ‘***’ represents *p* < 0.001.

**Table 2 foods-12-01844-t002:** Influence of general health interest, food neophobia scale, attitudes for alcoholic drinks, and attitudes for sweetness on liking, familiarity, and usage ratings for six fruit wine types in the online survey.

		General Health Interest (GHI)				Food Neophobia Scale (FNS)				Attitudes for Alcoholic Drinks				Attitudes for Sweetness		
		Liking	Familiarity	Usage		Liking	Familiarity	Usage		Liking	Familiarity	Usage		Liking	Familiarity	Usage
Grape	**G1**	3.58 ± 0.91	3.48 ± 1.03	2.15 ± 0.67	F1	3.68 ± 0.82	3.63 ± 0.88	2.25 ± 0.74	A1	3.26 ± 0.82 ^b^	3.29 ± 1.11 ^b^	1.93 ± 0.48 ^b^	S1	3.96 ± 0.84 ^a^	3.79 ± 0.81 ^a^	2.38 ± 0.71
	**G2**	3.70 ± 0.85	3.63 ± 0.97	2.31 ± 0.68	F2	3.63 ± 0.96	3.46 ± 1.08	2.24 ± 0.67	A2	3.71 ± 0.86 ^a^	3.53 ± 0.92 ^b^	2.16 ± 0.64 ^b^	S2	3.51 ± 0.89 ^b^	3.40 ± 1.03 ^b^	2.13 ± 0.67
					F3	3.63 ± 0.86	3.52 ± 1.06	2.16 ± 0.59	A3	3.88 ± 0.84 ^a^	3.77 ± 0.94 ^a^	2.53 ± 0.72 ^a^	S3	3.46 ± 0.82 ^b^	3.46 ± 1.10 ^b^	2.16 ± 0.61
Blueberry	**G1**	3.50 ± 0.83 ^b^	2.56 ± 0.98 ^b^	1.52 ± 0.56 ^b^	F1	3.63 ± 0.70	2.63 ± 0.97	1.52 ± 0.56	A1	3.45 ± 0.82	2.60 ± 1.02	1.45 ± 0.52 ^b^	S1	3.72 ± 0.83	2.78 ± 0.94	1.57 ± 0.62
	**G2**	3.75 ± 0.81 ^a^	2.89 ± 0.99 ^a^	1.69 ± 0.64 ^a^	F2	3.54 ± 0.93	2.72 ± 1.00	1.72 ± 0.64	A2	3.68 ± 0.88	2.80 ± 1.04	1.63 ± 0.62 ^b^	S2	3.56 ± 0.88	2.67 ± 0.95	1.60 ± 0.58
					F3	3.66 ± 0.84	2.78 ± 1.02	1.55 ± 0.57	A3	3.68 ± 0.76	2.71 ± 0.93	1.68 ± 0.62 ^a^	S3	3.57 ± 0.75	2.67 ± 1.09	1.61 ± 0.61
Hawthorn	**G1**	3.21 ± 1.00 ^b^	2.51 ± 1.09	1.44 ± 0.59	F1	3.30 ± 0.90	2.45 ± 1.01	1.42 ± 0.52	A1	3.22 ± 0.97	2.53 ± 1.08	1.40 ± 0.52	S1	3.43 ± 0.88	2.69 ± 1.02	1.41 ± 0.52
	**G2**	3.51 ± 0.90 ^a^	2.63 ± 1.01	1.46 ± 0.59	F2	3.33 ± 1.01	2.64 ± 0.99	1.57 ± 0.68	A2	3.42 ± 0.99	2.65 ± 0.99	1.52 ± 0.63	S2	3.32 ± 0.97	2.41 ± 1.00	1.44 ± 0.58
					F3	3.41 ± 0.93	2.60 ± 1.14	1.36 ± 0.53	A3	3.38 ± 0.93	2.50 ± 1.08	1.42 ± 0.58	S3	3.29 ± 1.04	2.59 ± 1.11	1.50 ± 0.64
Goji berry	**G1**	2.83 ± 0.88 ^b^	2.40 ± 1.03	1.33 ± 0.52	F1	3.03 ± 0.95	2.47 ± 1.10	1.40 ± 0.65	A1	2.73 ± 0.91	2.53 ± 1.06	1.22 ± 0.42 ^b^	S1	3.19 ± 0.92 ^a^	2.78 ± 1.03 ^a^	1.41 ± 0.52
	**G2**	3.07 ± 1.02 ^a^	2.68 ± 1.09	1.47 ± 0.66	F2	3.00 ± 0.98	2.58 ± 0.94	1.48 ± 0.60	A2	3.03 ± 0.96	2.58 ± 1.04	1.46 ± 0.63 ^a^	S2	2.82 ± 0.94 ^b^	2.34 ± 0.91 ^b^	1.36 ± 0.65
					F3	2.77 ± 0.90	2.54 ± 1.13	1.31 ± 0.49	A3	3.02 ± 0.96	2.47 ± 1.09	1.48 ± 0.65 ^a^	S3	2.83 ± 0.95 ^b^	2.49 ± 1.21 ^b^	1.41 ± 0.59
Apricot	**G1**	3.01 ± 0.87 ^b^	2.04 ± 0.85	1.25 ± 0.50	F1	3.17 ± 0.80	1.96 ± 0.77	1.25 ± 0.46	A1	2.94 ± 0.79 ^b^	1.87 ± 0.73 ^b^	1.14 ± 0.35 ^b^	S1	3.23 ± 0.85	2.08 ± 0.81	1.17 ± 0.42 ^b^
	**G2**	3.30 ± 0.79 ^a^	2.13 ± 0.83	1.31 ± 0.52	F2	3.13 ± 0.85	2.16 ± 0.82	1.30 ± 0.56	A2	3.08 ± 0.81 ^b^	2.10 ± 0.83 ^b^	1.27 ± 0.53 ^a^	S2	3.01 ± 0.81	2.03 ± 0.73	1.26 ± 0.49 ^a^
					F3	3.17 ± 0.88	2.13 ± 0.92	1.25 ± 0.49	A3	3.36 ± 0.87 ^a^	2.24 ± 0.90 ^a^	1.38 ± 0.30 ^a^	S3	3.19 ± 0.85	2.13 ± 0.97	1.37 ± 0.58 ^a^
*Rosa roxburghii*	**G1**	2.94 ± 0.84	1.92 ± 0.82	1.19 ± 0.41	F1	3.01 ± 0.79	1.87 ± 0.76	1.16 ± 0.40	A1	2.78 ± 0.77 ^b^	1.80 ± 0.64	1.09 ± 0.51	S1	3.08 ± 0.81	2.01 ± 0.75	1.12 ± 0.37
	**G2**	3.12 ± 0.85	2.11 ± 0.81	1.23 ± 0.49	F2	3.12 ± 0.82	2.13 ± 0.81	1.28 ± 0.50	A2	3.08 ± 0.86 ^b^	2.10 ± 0.83	1.25 ± 0.51	S2	2.98 ± 0.87	2.06 ± 0.81	1.25 ± 0.48
					F3	2.93 ± 0.91	2.04 ± 0.86	1.18 ± 0.42	A3	3.15 ± 0.86 ^a^	2.09 ± 0.90	1.26 ± 0.41	S3	3.00 ± 0.85	1.94 ± 0.88	1.24 ± 0.46

Groups were divided by TwoStep Cluster Analysis. GHI1 (Appendix A), the least health-interested (*n* = 126, GHI score ≤ 38); GHI2, the most health-interested (*n* = 108, GHI score > 38); FNS1, the least neophobics (*n* = 85, FNS score ≤ 23); FNS2 (*n* = 75, 23 < FNS score ≤ 26); FNS3, the most neophobics (*n* = 83, FNS score > 26); Alcohol 1, the negative attitude to alcohol (n = 71, Alcohol score ≤ 20); Alcohol 2 (*n* = 80, 20 < Alcohol score ≤ 25); Alcohol 3, the positive attitude (*n* = 83, Alcohol score > 25); Sweet 1, the least craving for sweet foods (*n* = 73, Sweet score ≤ 22); Sweet 2 (*n* = 84, 22 < Sweet score ≤ 27); Sweet 3, the most craving for sweet foods (*n* = 77, Sweet score > 27). Liking, Familiarity, and Usage score scales were 1–5. Letters ^a–b^ show significant differences (if detected) among subgroups within sample (ANOVA with Tukey’s, *p* < 0.05).

**Table 3 foods-12-01844-t003:** Concentrations (g/L) of sugars and organic acids in fruit wines.

	Blueberry Wines				Apricot Wine	Hawthorn Wines			*Rosa* *roxburghii* Wine	Goji Berry Wine
Samples	B1	B2	B3	B4	A	H1	H2	H3	R	G
**Glycerol**	6.9 ± 0.08 ^b,c^	5.9 ± 0.93 ^a,b,c^	14.4 ± 0.18 ^d^	1.2 ± 0.01 ^a,b^	7.5 ± 0.12 ^c^	n/d	4.2 ± 3.67 ^a,b,c^	3.5 ± 2.31 ^a,b,c^	5.8 ± 3.17 ^a,b,c^	3.8 ± 3.36 ^a,b,c^
**Fructose**	12.1 ± 0.10 ^b^	27.3 ± 0.14 ^c^	1.5 ± 0.05 ^a^	0.2 ± 0.02 ^a^	50.0 ± 0.51 ^e^	163.2 ± 6.27 ^f^	4.5 ± 0.03 ^a^	28.5 ± 0.37 ^c^	43.6 ± 1.54 ^d^	1.6 ± 0.03 ^a^
**Glucose**	7.8 ± 0.16 ^a,b^	24.1 ± 0.13 ^c^	n/d	n/d	21.1 ± 0.17 ^cd^	154.1 ± 8.37 ^d^	3.6 ± 0.03 ^a^	25.5 ± 0.16 ^c^	13.9 ± 0.41 ^b,c^	1.3 ± 0.02 ^a^
**Sorbitol**	n/d	n/d	n/d	n/d	2.1 ± 0.02 ^a^	10.6 ± 0.58 ^c^	6.5 ± 0.02 ^b^	n/d	n/d	n/d
**Sucrose**	1.6 ± 0.02 ^c^	3.5 ± 0.03 ^e^	n/d	n/d	0.4 ± 0.01 ^a^	1.2 ± 0.07 ^b^	13.4 ± 0.03 ^g^	4.2 ± 0.06 ^f^	n/d	1.8 ± 0.00 ^d^
**Total sugars**	28.4 ± 0.03 ^c,d^	60.8 ± 0.93 ^e^	15.9 ± 0.13 ^b,c^	1.4 ± 0.03 ^a^	81.2 ± 0.43 ^f^	329.1 ± 13.45 ^g^	32.3 ± 3.63 ^d^	61.8 ± 2.42 ^e^	63.3 ± 2.04 ^e^	8.5 ± 3.32 ^a,b^
**Succinic acid**	0.6 ± 0.03 ^c^	0.5 ± 0.03 ^b^	0.6 ± 0.01 ^d,e^	0.6 ± 0.01 ^e^	0.5 ± 0.03 ^b^	0.5 ± 0.01 ^b^	0.5 ± 0.01 ^b^	0.2 ± 0.03 ^a^	0.6 ± 0.02 ^c,d^	0.4 ± 0.02 ^b^
**Malic acid**	1.7 ± 0.00 ^f^	1.4 ± 0.02 ^e^	0.6 ± 0.00 ^d^	0.2 ± 0.01 ^b^	2.1 ± 0.02 ^g^	4.1 ± 0.05 ^h^	0.4 ± 0.02 ^c^	0.5 ± 0.01 ^d^	0.6 ± 0.01 ^d^	n/d
**Citric acid**	3.3 ± 0.07 ^d^	2.4 ± 0.03 ^c^	3.5 ± 0.02 ^d^	3.2 ± 0.04 ^d^	0.3 ± 0.17 ^a^	5.0 ± 0.49 ^e^	2.6 ± 0.01 ^c^	3.4 ± 0.06 ^d^	5.0 ± 0.11 ^e^	1.7 ± 0.03 ^b^
**Quinic acid**	8.6 ± 0.09 ^g^	7.1 ± 0.05 ^f^	2.7 ± 0.02 ^d^	0.7 ± 0.05 ^b^	7.0 ± 0.06 ^f^	n/d	1.4 ± 0.00 ^c^	4.8 ± 0.04 ^e^	1.3 ± 0.70 ^b,c^	n/d
**Galacturonic acid**	0.7 ± 0.04 ^a,b^	n/d	n/d	1.3 ± 0.11 ^a,b^	n/d	0.3 ± 0.23 ^a,b^	2.0 ± 1.24 ^b^	1.1 ± 1.39 ^a,b^	0.4 ± 0.09 ^a,b^	0.2 ± 0.16 ^a,b^
**Ascorbic acid**	n/d	n/d	n/d	n/d	n/d	n/d	n/d	n/d	1.7 ± 0.01 ^b^	n/d
**Total acids**	14.9 ± 0.05 ^d^	11.3 ± 0.06 ^c^	7.4 ± 0.04 ^b^	6.1 ± 0.10 ^b^	9.9 ± 0.22 ^c^	9.8 ± 0.28 ^c^	7.0 ± 1.22 ^b^	10.0 ± 1.39 ^c^	9.6 ± 0.69 ^c^	2.4 ± 0.19 ^a^
**Sugar/acid ratio**	1.9 ± 0.00 ^a,b^	5.4 ± 0.09 ^d,e^	2.1 ± 0.02 ^b^	0.2 ± 0.01 ^a^	8.2 ± 0.14 ^f^	33.5 ± 1.31 ^g^	4.7 ± 0.34 ^c,d^	6.2 ± 0.58 ^d,e^	6.6 ± 0.28 ^e,f^	3.5 ± 1.07 ^b,c^

All concentrations were mean values ± SD of three analytical replicates from triplicate samples. Samples with the same letters for each attribute do not differ significantly (One-way ANOVA and Tukey’s test, *p* < 0.05). n/d compound not detected.

## Data Availability

Data is contained within the article or Appendix A.

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
