# Peer review of "Exploring the Sensory Properties and Preferences of Fruit Wines Based on an Online Survey and Partial Projective Mapping"

_foods, 2023, doi:10.3390/foods12091844_

Round 1

Reviewer 1 Report

Dear Authors,

I thank the Editor for entrusting me to review this manuscript.

Below are my suggestions / comments:

The respondents to the survey were university students and employees (a non-representative sample).

A similar situation occurred with the sensory survey.

What criterion was used to select the number of clusters?

I understand that in Table 1 a comparison was made between users & non-users using ANOVA test, which is dedicated to more than 2 groups. A t-test is used to compare 2 groups.

Were the assumptions of the test checked before applying the ANOVA test?

All tables unreadable not very clear overloaded with information.

In tables 2 and 3 unreadable information on statistical comparisons.

Author Response

Point 1: The respondents to the survey were university students and employees (a non-representative sample).A similar situation occurred with the sensory survey.

Author's Response: Since the actual consumers of these fruit wines were still too few, and therefore difficult to recruit, we chose students and staff within the school. These populations were used instead of potential users that might have existed. Similarly, as in our previous study Lin et al. 2022 Food Research International. This current study is among the first studies aiming to characterize sensory quality of Chinese fruit wines, and the purpose of our study was not only consumer acceptance, but also in sensory quality and differences. With these factors in mind, we chose to use this so-called "convenience panel".

Point 2: What criterion was used to select the number of clusters?

Author's Response: When GHI was divided into three groups, there was no significant difference in their Liking, Familiarity and Usage, which was lack of statistical significance. We tried to divide them into high GHI and low GHI populations (Just like this article: https://doi.org/10.3390/foods10010060) and more significant differences emerged. Our goal was to compare the differences between the populations, not simply to divide them into three groups.

Point 3: I understand that in Table 1 a comparison was made between users & non-users using ANOVA test, which is dedicated to more than 2 groups. A t-test is used to compare 2 groups.

Author's Response: Thank you for the correction, which has been made in Table 1 and in the text.

Point 4: Were the assumptions of the test checked before applying the ANOVA test?

Author's Response: Yes, the normality of the distributions were tested. The independence test and chi-square test have been passed, but when we tested normal distribution, the data were perfectly normally distributed. However, still sufficiently in order to use parametric test ANOVA.  

Point 5: All tables unreadable not very clear overloaded with information.In tables 2 and 3 unreadable information on statistical comparisons.

Author's Response: The tables and figures in the peer-reviewed manuscript were narrowed, which did not match our original icon format, and we have contacted the editor to revise them. We have submitted the original tables again.

Reviewer 2 Report

The work submitted to Foods “Exploring the sensory properties and preferences of fruit wines based on an online survey and partial projective mapping” aims to explore young consumers’ fruit wine preferences and descriptors for the varied fruit wines.

There are already several works published about Projective Mapping:

Luisa Torri, Caterina Dinnella, Annamaria Recchia, Tormod Naes, Hely Tuorila, Erminio Monteleone, Projective Mapping for interpreting wine aroma differences as perceived by naïve and experienced assessors, Food Quality and Preference, Volume 29, Issue 1, 2013, Pages 6-15, https://doi.org/10.1016/j.foodqual.2013.01.006.

Wilson C, Brand J, du Toit W, Buica A. Matrix effects influencing the perception of 3-mercaptohexan-1-ol (3MH) and 3-mercaptohexyl acetate (3MHA) in different Chenin Blanc wines by Projective Mapping (PM) with Ultra Flash profiling (UFP) intensity ratings.. Food Res Int. 2019 Jul;121:633-640. doi: 10.1016/j.foodres.2018.12.032

Hayward L, McSweeney MB.Investigating caloric values and consumers' perceptions of Nova Scotia rosé wines. Food Res Int. 2020 Jan;127:108761. doi: 10.1016/j.foodres.2019.108761

Wong B, Muchangi K, Quach E, Chen T, Owens A, Otter D, Phillips M, Kam R.. Characterisation of Korean rice wine (makgeolli) prepared by different processing methods. Curr Res Food Sci. 2022 Dec 27;6:100420. doi: 10.1016/j.crfs.2022.100420.

The present work introduces an interesting application of this technique – Fruit wines.

The work is well planned and executed. The results are interesting and lead to formal conclusions.

I have just found one minor mistake:

Line 57 – The reference is incomplete Parpinello et al.,

Figure 1 and Figure 2 should also be enlarged. It is no possible to see!

Author Response

Thank you for your references!

Point 1: Line 57 – The reference is incomplete Parpinello et al.,

Author's Response: Thank you for the correction, this has been revised.

Point 2: Figure 1 and Figure 2 should also be enlarged. It is no possible to see!

Author's Response: We apologize that the tables and figures in the peer-reviewed manuscript were narrowed, which did not match our original icon format, and we have contacted the editor to revise them.We have also changed the layout of the figures, they are now larger.

Reviewer 3 Report

The Authors chose a topic of fruit wines made of several species. In their study, they combined an online survey with a partial projective mapping procedure. From lines 56, the Authors explain the healthiness of food and beverages in relation to consumer choices. It would be important to note here, that the regulation on health claims is relatively strict in many markets. Of course, a general education of consumers about health related, or beneficial compounds is desirable, however in many markets it is not allowed to make any health claims on product labels, only in case when someone takes the whole process of legalization. So a short sentence about communicating health claims with consumers should be involved in this section.

The questionnaire and the sensory methodology is good. The Authors have followed the elements of good sensory practice, with the coding of the samples, using standardized tasting glasses and providing palate cleansers during the session. Compusense is a proven software for such kind of data collection.

In line 220 the Authors describe, that they have used cluster analysis to classify respondents on the basis of their consumption frequency. It would be interesting to perform such kind of cluster analysis, when all relevant responses are included in the data table, and they run a ‘global’ clustering based on their responses. But the approach they have used is also acceptable.

The study is effectively combining the different approaches (questionnaires, projective mapping data and measurements of liking). It is valuable study for those, who are also interested in making similar studies in the future.

Author Response

Point 1: From lines 56, the Authors explain the healthiness of food and beverages in relation to consumer choices. It would be important to note here, that the regulation on health claims is relatively strict in many markets. Of course, a general education of consumers about health related, or beneficial compounds is desirable, however in many markets it is not allowed to make any health claims on product labels, only in case when someone takes the whole process of legalization. So a short sentence about communicating health claims with consumers should be involved in this section.

Author's Response: Thank you for your advice! Wine is unhealthy and we have emphasized the health benefits of these berry ingredients.

Reviewer 4 Report

After reading the manuscript " Exploring the sensory properties and preferences of fruit wines  based on an online survey and partial projective mapping", I realized that the manuscript showed in some parts the scientific rigour wanted, but in other parts I have missed it.

The authors have presented critical evaluation only in some paragraphs.

The references are not exactly current.

Thats why I have written some suggestions below in an attempt to improve the paper.

L.1- Suggestion based on your objective : Exploring consumers recognition and preferences of sensory characteristics of fruit wines based on a survey and  mapping

It is not necessary to give details in the title of what is in the Material and methods.

L.12 - I wouldn't use "young".

40 years old - we are already mature, vivid, experienced, active population ... very different from 18 years old.

Think about it. Check the whole paper.

L.13- Your abstract objective does not seem to me to cover all your paper. Remember that many researchers discontinue reading or read some paper after reading only the abstract. 

L.28- Do not use in your keywords, words that are already in your title. This reduces the chances of other researchers finding your paper. I suggest the fruit, scientific names : blueberry, apricot , hawthorn, hawthorn, (Rosa roxburghii  and goji .

L. 36- Please, insert something about world consumption as well, not just about China. This will attract the attention of readers from other countries.

L-40 - What about this world wide dominance?  Which country would be the largest consumer?

L.57-  Parpinello et al., - Attention to Journal rules

L.60- I think it is important to include other examples to enrich the point about health, resveratrol who knows.

L.64- " it is still a relatively small product category," -  So I recommend inserting some country that has successful experience in order to see both sides.

L.65 - It became repetitive, often "Chinese". Although the Chinese fruit wine industry has experienced decades of development since its establishment, it is still a relatively small product category, e.g., relatively novel drinks for  Chinese consumers in the broad Chinese market. Therefore, it is valuable to investigate whether consumers' consumption of fruit wine is affected by food neophobia or other attitudinal features.

L.72- between or among ?

L.94- Here we have some vulnerability. More information about the post-harvest period has to be inserted: ripeness point of all the fruits ? how were they transported ? refrigerated ? and from the arrival to the use in the research how were they conditioned, including the temperature ?  How were they selected and homogenized to be used in the research ? How many replications ? Was the actual fruit or the pulp used?

L.100-  Did the paper follow the Helsinki declaration? Please, enter the approval protocol number.

L.103  and 136- Repetitive : " and their rights to  cancel participation." and "  In addition, they were informed that they could cancel their participa- 136 tion at any time"

L.144- Were the analyses performed in sensory booths ?

L.220- What would be "Users" ? If you considered "non users" for the evaluators who did not have the habit of consuming wine, we have a flawn in the study, because these people should not participate. The term "user" is not common in sensory analysis.  We have: assessor, evaluator, taster. Please, review it. Sorry, but I did not understand.

L.241- Tabel 1  you don't need to have "wine" in all columns, because you have already written in the upper cell " Fruit wine type". You could reduce the font and fix the standard deviations on the same line, it was too unconfigured.

L.267 -Table 2-  you don't need to have "wine" in all columns, because you have already written in the upper cell " Fruit wine type".

Make table 2 clearer about what would be  1, 2, 3. Maybe A1, A2, A,3 and S1,S2,S3.  The table needs to be self-explanatory. 

Table 2- You could reduce the font and fix the standard deviations on the same line, it was too unconfigured.

L.282 and 282- Between or among. Check the whole paper, please.

L.296- " first two dimensions explaining 47.04% and 15.82% of the total data variance." - I missed a discussion of what might have happened.

L.321- Table 3-  I am really sorry, but it is really hard to accept this table this way. My suggestion would be to avoid repeating wine, maybe using acronyms. For example : Dry (D) and so on. In the "raw material" line,  do you really need to write the name of the fruit again? This line in my opinion should be the first line and not the third. Your tables are very disconfigured, it is difficult even for us to evaluate them. I don't know if it looks like this way to you, but the standard deviations are in the bottom row, piled up with the averages.

L.346, 359, 362, 515 ... and conclusion-  comparing untrained X trained and 

" The result of this study found that un-trained consumers could distinguish different fruit wines based on their sensory characteristics."

L.399 and L.403 - Figure 2 and Figure 3 - I did not understant, something is missing.

L.484 and L484- Follow Journals Guideline about authors -  Golia et al. Honoré-Chedozeau et al.

L.515- 531-  Your conclusion did not seem to me to contemplate all the findings of your paper. Please, reevaluate

- Was it your objective as well ? Check your objective. 

L.418 - ". Ristic et al." Follow Journal's Guideline.

Author Response

L.1- Suggestion based on your objective: Exploring consumers recognition and preferences of sensory characteristics of fruit wines based on a survey and mapping. It is not necessary to give details in the title of what is in the Material and methods.

Author's Response: Thank you for your suggestion. However, we consider such a title to bring more useful and specified information to the potential reader.

L.12 - I wouldn't use "young". 40 years old - we are already mature, vivid, experienced, active population ... very different from 18 years old. Think about it. Check the whole paper.

Author's Response: Thank you for your suggestion. We have revised it to "adult participants".

L.13- Your abstract objective does not seem to me to cover all your paper. Remember that many researchers discontinue reading or read some paper after reading only the abstract. 

Author's Response: Thanks for your advice! We have revised it to include more of the contents: This study aimed to explore young consumers’ fruit wine preferences and descriptors for the varied fruit wines, and investigate the impact of general heath interest, food neophobia scale, attitudes towards sweets and attitudes towards alcohol on consumers.

L.28- Do not use in your keywords, words that are already in your title. This reduces the chances of other researchers finding your paper. I suggest the fruit, scientific names: shus

Author's Response: We have modified some of the keywords. However, we do not think we should use the names of the berries because they happen to be current ones due to their availability at the time of conducting the study. The ingredients in these wines are locally important and they are not widely available on markets.

L.36- Please, insert something about world consumption as well, not just about China. This will attract the attention of readers from other countries.

Author's Response: Consumption of berry-based wines is still low worldwide but constantly increasing with interests to use locally important ingredients in fruit wine making, and we have added related content.

L-40 - What about this world wide dominance?  Which country would be the largest consumer?

Author's Response: The markets are still significantly smaller in comparison to many other alcoholic beverage markets. Thus there are enough publish studies or statistics about the consumption being done right now. More research is needed to support the future development of the fruit wine market. Our study also aims to promote fruit wines, in addition to the specific scientific aims. More on: https://doi.org/10.1016/B978-0-12-800850-8.00007-7

L.57-  Parpinello et al., - Attention to Journal rules

Author's Response: This has been revised.

L.60- I think it is important to include other examples to enrich the point about health, resveratrol who knows.

Author's Response: We have edited this part to further emphasize berry ingredients.

L.64- " it is still a relatively small product category," -  So I recommend inserting some country that has successful experience in order to see both sides.

Author's Response: Currently, there are no countries with real successful experiences in this area. However, the utilization of locally important ingredients in different product categories (including fruit wines and other alcoholic, low-alcoholic and non-alcoholic beverages) is constantly increasing in China and Finland, as well as in other countries in the Northern Europe (e.g Norway and Estonia). 

L.65 - It became repetitive, often "Chinese". Although the Chinese fruit wine industry has experienced decades of development since its establishment, it is still a relatively small product category, e.g., relatively novel drinks for  Chinese consumers in the broad Chinese market. Therefore, it is valuable to investigate whether consumers' consumption of fruit wine is affected by food neophobia or other attitudinal features.

Author's Response: We have made the deletions and revised this part.

L.72- between or among ?

Author's Response: Thank you for the careful examination! It should be 'among'.

L.94- Here we have some vulnerability. More information about the post-harvest period has to be inserted: ripeness point of all the fruits ? how were they transported ? refrigerated ? and from the arrival to the use in the research how were they conditioned, including the temperature ?  How were they selected and homogenized to be used in the research ? How many replications ? Was the actual fruit or the pulp used?

Author's Response: The samples used in this study were commercially available fruit wines. We do not have details how they were processed.

L.100-  Did the paper follow the Helsinki declaration? Please, enter the approval protocol number.

Author's Response: Our experiments were conducted following the Helsinki Declaration. To be more specific, we followed the guidelines used in Finland (https://tenk.fi/en) for non-medical research involving human participants. We informed participants about the aims of the study and the ethical issues related to the study, and also their rights as panelists to cancel participation and ask more information.

However, the data for this study was collected before the provision for ethical pre-evaluation was enacted in 2019 (https://tenk.fi/en/ethical-review). And no similar regulations have been issued in China yet. Therefore, we do not have ethical pre-evaluation statement or number.

L.103  and 136- Repetitive : " and their rights to  cancel participation." and "  In addition, they were informed that they could cancel their participa- 136 tion at any time"

Author's Response: Thank you for checking, I have removed the content of L103.

L.144- Were the analyses performed in sensory booths ?

Author's Response: Participants had separate tables to conduct the test whereas there were dividers or walls between the tables. All participants were instructed not to disturb others while still being able to ask more information about the study. 

L.220- What would be "Users" ? If you considered "non users" for the evaluators who did not have the habit of consuming wine, we have a flawn in the study, because these people should not participate. The term "user" is not common in sensory analysis.  We have: assessor, evaluator, taster. Please, review it. Sorry, but I did not understand.

Author's Response: Currently, there are too few real consumers of these fruit wines available for scientific studies thus we were not able to find sufficient number of ‘real’ consumers." Users" and "Non-users" are not actual consumers/non-consumers, they are two groups that we have named in this study by clustering the frequency of consumption. In short, they are two data labels.

L.241- Tabel 1  you don't need to have "wine" in all columns, because you have already written in the upper cell " Fruit wine type". You could reduce the font and fix the standard deviations on the same line, it was too unconfigured.

Author's Response: Thank you for the suggestion, we have removed 'wine'. We apologize that the tables and figures in the peer-reviewed manuscript were narrowed, which did not match our original icon format, and we have contacted the editor to revise them.

L.267 -Table 2-  you don't need to have "wine" in all columns, because you have already written in the upper cell " Fruit wine type".

Author's Response: Thank you for the suggestion, we have removed 'wine'. We also modified the width of Table 2 to make it easier to read.

Make table 2 clearer about what would be  1, 2, 3. Maybe A1, A2, A,3 and S1,S2,S3.  The table needs to be self-explanatory. 

Author's Response: You are right, I added abbreviations in front of 1, 2, 3 in the table.

Table 2- You could reduce the font and fix the standard deviations on the same line, it was too unconfigured.

Author's Response: Thank you for the suggestion, we modified the width of Table 2 to make it easier to read.

L.282 and 282- Between or among. Check the whole paper, please.

Author's Response: Thanks for the careful check, I have checked all the usage of between in the whole text and made the changes.

L.296- " first two dimensions explaining 47.04% and 15.82% of the total data variance." - I missed a discussion of what might have happened.

Author's Response: Thank you for the check, we revised this section.

L.321- Table 3-  I am really sorry, but it is really hard to accept this table this way. My suggestion would be to avoid repeating wine, maybe using acronyms. For example : Dry (D) and so on. In the "raw material" line,  do you really need to write the name of the fruit again? This line in my opinion should be the first line and not the third. Your tables are very disconfigured, it is difficult even for us to evaluate them. I don't know if it looks like this way to you, but the standard deviations are in the bottom row, piled up with the averages.

Author's Response: Thank you for your advice! We moved the first three rows to Table S1 so that Table 3 becomes concise.

L.346, 359, 362, 515 ... and conclusion-  comparing untrained X trained and 

" The result of this study found that un-trained consumers could distinguish different fruit wines based on their sensory characteristics."

Author's Response: We collected descriptors from these untrained consumers, and they were able to distinguish between the samples. However, they provided many vague attributes, such as "balanced fragrance", "unbalanced fragrance". Future studies are needed to define the key sensory attributes using more trained sensory panels.

L.399 and L.403 - Figure 2 and Figure 3 - I did not understant, something is missing.

Author's Response: We are very sorry that Figure 2 was lost in the editor's layout, we are communicating. We will provide new figures.

L.484 and L484- Follow Journals Guideline about authors -  Golia et al. Honoré-Chedozeau et al.

Author's Response: These have been revised.

L.515- 531-  Your conclusion did not seem to me to contemplate all the findings of your paper. Please, reevaluate

- Was it your objective as well ? Check your objective. 

Author's Response: Thank you, we have made the additions to the conclusions.

L.418 - ". Ristic et al." Follow Journal's Guideline.

Author's Response: This has been revised.

Round 2

Reviewer 4 Report

Dear authors, 

The authors have accepted some suggestions that I gave in relation to the 1st version, but for me the paper has many  flaws in methodology and  besides  the products have no worldwide appeal  yet.

Thepapers that I have read or given advises  for Foods have  well-designed  figures, tables,  flowcharts, schemas .... to give a differentiated approach and attract the readers/other researchers.  In this paper this was not well done, and I suggested it previously.

In the first version I already had reservations, but since the subject seemed important... I expected a progress in this second version, which unfortunately did not occur.

I can not accept it,  so sorry. 

Author Response

Sorry, we can't agree that the "product has no global appeal". Berries and fruits are harvested (cultivated or wild) globally and their short shelf life can lead to huge waste. These fruit wines are a novel attempt to reduce waste, and we hope that these products will be accepted.

And, we have re-edited the discussion to emphasize the PM portion of the study and its usage/limitations as a test method, and to place less emphasis on online survey due to the limitations of our participants and "users". We hope you will re-read them, and thank you very much!